# Molecular Cloning and Exploration of the Biochemical and Functional Analysis of Recombinant Glucose-6-Phosphate Dehydrogenase from *Gluconoacetobacter diazotrophicus* PAL5

**DOI:** 10.3390/ijms20215279

**Published:** 2019-10-24

**Authors:** Edson Jiovany Ramírez-Nava, Daniel Ortega-Cuellar, Abigail González-Valdez, Rosa Angélica Castillo-Rodríguez, Gabriel Yaxal Ponce-Soto, Beatriz Hernández-Ochoa, Noemí Cárdenas-Rodríguez, Víctor Martínez-Rosas, Laura Morales-Luna, Hugo Serrano-Posada, Edgar Sierra-Palacios, Roberto Arreguin-Espinosa, Miguel Cuevas-Cruz, Luz María Rocha-Ramírez, Verónica Pérez de la Cruz, Jaime Marcial-Quino, Saúl Gómez-Manzo

**Affiliations:** 1Laboratorio de Bioquímica Genética, Instituto Nacional de Pediatría, Secretaría de Salud, Ciudad de México 04530, Mexico; edsonjiovany@ciencias.unam.mx (E.J.R.-N.); ing_vicmr@hotmail.com (V.M.-R.); lauraeloisamorales@ciencias.unam.mx (L.M.-L.); 2Laboratorio de Nutrición Experimental, Instituto Nacional de Pediatría, Secretaría de Salud 04530, Mexico; dortegadan@gmail.com; 3Departamento de Biología Molecular y Biotecnología, Instituto de Investigaciones Biomédicas, Universidad Nacional Autónoma de México, Ciudad de Mexico 04510, Mexico; abigaila@biomedicas.unam.mx; 4Consejo Nacional de Ciencia y Tecnología (CONACYT), Instituto Nacional de Pediatría, Secretaría de Salud, Ciudad de Mexico 04530, Mexico; racastilloro@conacyt.mx (R.A.C.-R.); jmarcialq@ciencias.unam.mx (J.M.-Q.); 5Institute for Bio- and Geosciences (IBG-2: Plant Sciences), Forschungszentrum Jülich, Wilhelm Johnen Straße, 52428 Jülich, Germany; yaxal@iecologia.unam.mx; 6Laboratorio de Inmunoquímica, Hospital Infantil de México Federico Gómez, Secretaría de Salud, Ciudad de Mexico 06720, Mexico; beatrizhb_16@comunidad.unam.mx; 7Laboratorio de Neurociencias, Instituto Nacional de Pediatría, Secretaría de Salud, Ciudad de México 04530, Mexico; noemicr2001@yahoo.com.mx; 8Consejo Nacional de Ciencia y Tecnología (CONACYT), Laboratorio de Agrobiotecnología, Tecnoparque CLQ, Universidad de Colima, Carretera los Limones-Loma de Juárez, Colima 28629, Mexico; hjserranopo@conacyt.mx; 9Colegio de Ciencias y Humanidades, Plantel Casa Libertad, Universidad Autónoma de la Ciudad de México, Ciudad de Mexico 09620, Mexico; edgar.sierra@uacm.edu.mx; 10Departamento de Química de Biomacromoléculas, Instituto de Química, Universidad Nacional Autónoma de Mexico, Ciudad de Mexico 04510, Mexico; arrespin@unam.mx (R.A.-E.); miguel.ccqi@yahoo.com.mx (M.C.-C.); 11Departamento de Infectología, Hospital Infantil de México Federico Gómez, Dr. Márquez No. 162, Col Doctores, Delegación Cuauhtémoc 06720, Mexico; luzmrr7@yahoo.com.mx; 12Departamento de Neuroquímica, Instituto Nacional de Neurología y Neurocirugía Manuel Velasco Suárez, S.S.A., Ciudad de Mexico 14269, Mexico; veped@yahoo.com.mx

**Keywords:** glucose 6 phosphate dehydrogenase (G6PD), bioinformatics analysis, kinetic parameters, thermal stability, three-dimensional structure

## Abstract

*Gluconacetobacter diazotrophicus* PAL5 (GDI) is an endophytic bacterium with potential biotechnological applications in industry and agronomy. The recent description of its complete genome and its principal metabolic enzymes suggests that glucose metabolism is accomplished through the pentose phosphate pathway (PPP); however, the enzymes participating in this pathway have not yet been characterized in detail. The objective of the present work was to clone, purify, and biochemically and physicochemically characterize glucose-6-phosphate dehydrogenase (G6PD) from GDI. The gene was cloned and expressed as a tagged protein in *E*. *coli* to be purified by affinity chromatography. The native state of the G6PD protein in the solution was found to be a tetramer with optimal activity at pH 8.8 and a temperature between 37 and 50 °C. The apparent *Km* values for G6P and nicotinamide adenine dinucleotide phosphate (NADP^+^) were 63 and 7.2 μM, respectively. Finally, from the amino acid sequence a three-dimensional (3D) model was obtained, which allowed the arrangement of the amino acids involved in the catalytic activity, which are conserved (RIDHYLGKE, GxGGDLT, and EKPxG) with those of other species, to be identified. This characterization of the enzyme could help to identify new environmental conditions for the knowledge of the plant–microorganism interactions and a better use of GDI in new technological applications.

## 1. Introduction

*Gluconacetobacter diazotrophicus* PAL5 (GDI) is a strict aerobe and a nitrogen-fixing acetic acid bacterium that was originally isolated from sugar cane [1]. It grows under stringent conditions, such as in areas with a high sucrose content and low pH [2]. GDI is an excellent model system for the study of plant–microbe interactions [3,4], since it performs symbiosis with different plants, such as potato, sorghum, coffee, and pineapple [5,6], by colonizing their stems, leaves, and the intracellular space of sugarcane roots [1].

Whole genome sequencing allows for better understanding of the GDI, and the knowledge of several molecular and biochemical aspects of this microorganism have opened new research directions. The GDI genome is composed of one chromosome (3.9 Mb) and two plasmids (16.6 and 38.8 kb, respectively). About 3938 coding sequences have been annotated, including those related to sugar metabolism, such as that of the pentose phosphate pathway (PPP) [7,8].

An analysis of carbon and nitrogen metabolism for GDI showed that glucose is the principal carbon source. Accordingly, the first step for glucose oxidation to gluconate is catalysis by membrane-bound quino-protein glucose dehydrogenases; however, in situations with excess glucose, a second enzyme, named NAD-GDH (GDI2625), also promotes its oxidation [9]. Furthermore, it has been proposed that complete glucose oxidation may occur through several enzymes such as protein kinase (GDI3115), 2-ketogluconate reductase (GDI3432), and 6-phosphogluconate dehydrogenase-NAD, which drive the movement of gluconate to the PPP [9], since the key enzymes of the Embden–Meyerhof–Parnas (6-phosphofructokinase and 1-phosphofructokinase) and Entner–Doudoroff (6-phosphogluconate dehydratase and 2-keto-3-deoxyphosphogluconate aldolase) systems have not been detected [1,10,11].

Given the relevance of the function of the PPP, whose key regulatory enzyme is glucose-6-phosphate dehydrogenase (G6PD) in the glucose metabolism of GDI, in this work, for the first time we report the cloning and heterologous expression of complete active recombinant G6PD from GDI. This study allowed us to analyze the detailed steady state kinetics, thermostability, and biophysical characterization of the purified G6PD enzyme. Finally, using homologous 3D structures, we modeled the G6PD protein, which suggests the structural and functional features of the G6PD from several organisms, including humans.

## 2. Results and Discussion

### 2.1. zwf cDNA Isolation and Characterization

The *zwf* gene was obtained by PCR using specific primers and then cloned into the vector pJET 1.2. to analyze the sequence. The clone obtained contained an open reading frame of 1530 bp in length, which predicted a 510 amino acid protein with a theoretical molecular weight (MW) of 56,483 Da and an isoelectric point (pI) of 5.94, which was determined with the ProtParam tool from the Expasy online program [12]. The cloned *zwf* sequence analyzed by blast nucleotide (BLASTn) sequencing in GenBank found a 100% similarity with the nucleotide sequence of G6PD from GDI (GenBank: CAP54231.1) [13]. Then, the nucleotide sequence obtained was translated into an amino acid sequence. A comparison of the *G. diazotrophicus* PAL5 G6PD amino acid sequence was performed with HHblits [14] and revealed 100% levels of similarity with the putative G6PD from *Tanticharoenia sakaeratensis* (accession number: A0A0D6ML00), *Verrucomicrobiae bacterium* (accession number: A0A2A2YE23), and *Xanthomonas cucurbitae* (accession number: A0A2S7DNS6), among others (Appendix A).

Then, 60 amino acid sequences of G6PDs from different species were retrieved from Swiss-Prot and NCBI (48 sequences from Swiss Prot and 12 sequences from NCBI) (Appendix A), and a multiple alignment was performed with MAFFT V7.427 [15] (zwf_mafft.aln.fasta) and visualized with the online program Jalview [16] in order to find G6PD orthologs in GDI. As a result of the alignment, three conserved fragments were identified, which were previously characterized in human G6PD and are present in G6PD from GDI (Appendix A). The first fragment corresponds to the amino acids 31-GxGGDLT-37 (amino acid number corresponding to the G6PD sequence from GDI). This fragment, previously reported for human G6PD, is involved in the binding of catalytic nicotinamide adenine dinucleotide phosphate (NADP^+^ coenzyme). The second conserved fragment, 169-EKPxG-173 (amino acid number corresponding to G6PD sequence from GDI), contains proline 171 (Pro171), which is related to the correct positioning of the substrate (G6P) and coenzyme (NADP^+^) during enzymatic reactions [17]. Finally, the third fragment, 197-RIDHYLGKE-205 (amino acid number corresponding to G6PD sequence from GDI), which has nine amino acids, contains lysine (Lys204), which is responsible for the binding of the substrate in human G6PD [18]. Furthermore, this amino acid sequence contains aspartate, histidine, and lysine, which are considered important for the binding of the G6P substrate in G6PD from *L. mesenteroides* [19].

To determine the evolutionary relationship between the *G. diazotrophicus* G6PD protein and the other G6PD homologs, we constructed a phylogenetic tree (Appendix A) (zwf_mafft.tree). Phylogenetic analysis revealed that the G6PD sequence from GDI is most closely related to the G6PD sequences found in organisms like *Gluconacetobacter liquefaciens* and *Gluconacetobacter etanii*, as well as species of genus *Acetobacter* (Appendix A).

### 2.2. Heterologous Expression and Purification of *G6PD*

To better characterize G6PD from GDI, the pJET 1.2 vector containing the *zwf* gene was digested, sub cloned, and overexpressed in the pET3a-HISTEVP plasmid under the conditions described in the Materials and Methods section. The G6PD protein, which is fused to an N-terminal His tag, was purified by Ni Sepharose high performance affinity column (GE Healthcare). Then, the tag was removed using the site-specific protease HisTEVP. The inset in Figure 1 shows that, after SDS-PAGE, the protein was detected as a single band with an apparent MW of 56.4 kDa. This result is consistent with the theoretical MW of the monomeric form of G6PD. Of note, this size is similar to those reported for G6PD proteins in other prokaryotes as *Thermotoga maritima* and *Pseudomonas aeruginosa* [20,21], but it is different for the organs of some mammals, such as the camel liver, rat kidney, and buffalo liver, since they have MWs of around 60 to 75 kDa [22,23,24].

Furthermore, the native status of the G6PD protein was determined by size exclusion chromatography (Figure 1A). As shown in the chromatogram, a single peak with an elution volume of 44.4 mL with G6PD activity was observed. From the calibration curve plot, we found that the protein mass corresponds to the native tetramer (Figure 1B) with a MW of 220 ± 5 kDa, which is in accordance with the MW expected from the amino acid sequence (56.4 kDa × 4 ≈ 225 kDa). In addition, we found a second small peak with a relative MW of around 120 kDa, which could be a dimeric form, but no G6PD activity was observed for this form (Figure 1A). Furthermore, no monomer MW or larger aggregates were observed in the fast protein liquid chromatography (FPLC) chromatogram. These results are in accordance with those previously reported, where a tetrameric structure for G6PDs from *Brugia malayi* and *Pseudomonas aeruginosa* were reported [21,25]. However, the native statuses of G6PDs from camel liver, *L. mesenteroides*, and *Haloferax volcanii* have been reported as dimers and trimers, respectively [22,26,27].

### 2.3. Biochemical Analysis of the *G6PD* Protein

#### 2.3.1. Effect of pH and Temperature on G6PD Activity

To determine the effect of pH and temperature on the activity of the G6PD enzyme, we examined the activity at different pH values and temperatures. The activity of G6PD was measured at pH values ranging from 2.6 to 10.0, with an optimal pH noted at 8.2. As can be observed at pH values from 2.6 to 4.0, no G6PD activity was detected; at a pH value of 5.0, G6PD activity was found at around 10%. Furthermore, we observed a relative activity of 42% at pH 7.0, while at pH 10.0, a residual activity of 60% was observed (Figure 2A). We suggest that this enzyme, which has an optimal pH of 8.2, probably participates during the growth phase, where the high demands of precursors, such as NADPH (notably required for lipid biosynthesis, cholesterol biosynthesis, and protection against cellular oxidative stress) or ribose are provided by the PPP. Furthermore, this result agrees with those previously reported, where the enzyme had a similar pH profile compared with the other previously purified G6PDs. For example, pH values of around 7.5 to 8 were found in G6PDs from *Homo sapiens*, *P. aeruginosa*, *B. malayi*, buffalo liver, camel liver, dog liver, *T. crassiceps*, *T. cruzy*, *T. maritima*, *E. coli* DH5α, and *A. Oryzae* [20,21,22,24,25,28,29,30,31,32,33]. However, an optimal pH value of around 9 has also been reported for G6PDs from *A. niger* and *A. midulms* [34].

To investigate the effect of temperature on the activity of G6PD protein, aliquots of the enzyme were incubated at different temperatures (ranging from 15 to 60 °C), and, subsequently, the activity was measured. As seen in Figure 2B, the activity remained intact from 37 to 50 °C and slowly decreased between 47 and 52 °C, and then sharply dropped. Furthermore, we observed that the G6PD enzyme loses about 60% of its activity when incubated at temperatures below 30 °C. The stability of the G6PD enzyme from GDI is in agreement with that previously reported for G6PDs from *A. Orizae* and *Homo sapiens* [33,35,36], as it has been observed that these enzymes have no alteration in their enzymatic activity from 35 to 50 °C. However, the G6PD enzymes from *B. malayi* and *Y. lipolytica* [25,37] lost 100% of their activity at 50 °C, while the G6PD of GDI showed 100% activity at this temperature, which indicates that this protein is more stable under high temperatures.

#### 2.3.2. Kinetic Study of the G6PD Enzyme

The steady-state kinetic parameters of G6PD were obtained by varying the concentration of one of the substrates (G6P or NADP^+^) while keeping the other (G6P or NADP^+^) fixed at ∼10-fold of the Km value. As seen in Figure 3, hyperbolic behavior of the G6PD enzyme was observed for both substrates, G6P and NADP^+^, and the initial velocity values were fitted to the Michaelis–Menten equation by non-linear regression calculations. The apparent K_m_ values for G6P and NADP^+^ were 63 and 7.2 μM respectively (Table 1), with an apparent *V*_max_ of 43 µmol·min^−1^·mg^−1^. These results are similar to those reported for G6PDs from different organisms, where the K_m_ values for NADP^+^ ranged from 6 to 10 μM [21,29,36], but the K_m_ for G6PD is lower than previously reported values (Table 1) [20,21,34]. In addition, as can be observed in Table 1, the G6PD protein has a higher catalytic constant (k_cat_) value (293,181 s^−1^) compared to that of human G6PD (233 s^−1^), *P. aeruginosa* (540 s^−1^), *A. orizae* (1000 s^−1^), and *T. maritima* (35,000 s^−1^) [20,21,33].

### 2.4. Evaluation of Protein Stability

#### 2.4.1. Thermal Inactivation Analysis

Thermal inactivation assays have been widely used to evaluate the stability of the active site of the human WT G6PD enzyme and its variants [35,36,40,41,42,43,44,45,46]. As shown in Figure 4A, when the enzyme was incubated with 1 K_m_ (7.2 µM) of NADP^+^, no protective effect on thermal stability was observed, since its temperature *T*_50_ (temperature at which the enzyme loses 50%) remained similar with or without NADP^+^. This result is in disagreement with that previously reported for recombinant human G6PD, whose *T*_50_ increases from 47 to 59 °C in the presence of NADP^+^ [46]. This probably indicates that recombinant G6PD does not have the amino acids required for the binding of the structural NADP^+^ molecule, as occurs in eukaryotes, such as the human G6PD enzyme. It is interesting to note that in the alignment of the amino acid sequence of the G6PD from GDI, this enzyme contain only 1 amino acid R376 (amino acid number corresponding to G6PD sequence from GDI) of the 10 amino acids that have been reported to participate in the binding of structural NADP^+^ in human G6PD [17,47].

In addition, because we did not find a protective effect of G6PD in the presence of NADP^+^, we performed thermal inactivation assays in the presence or absence of G6P (second physiological substrate). This condition is closer to its physiological habitat, because GDI is grown in a large part of the intracellular space of sugarcane roots, where high concentrations of glucose are present in the environment. As seen in Figure 4A, we noted no protective effect when the enzyme was incubated with 1 K_m_ (63 µM) of G6P. A *T*_50_ of 50 °C was determined both in the absence and presence of 1 K_m_ of G6P This result indicates that the G6PD is not protected by NADP^+^ or G6P.

#### 2.4.2. Stability of G6PD in the Presence of Guanidine Hydrochloride (Gdn-HCl) or Protease Digestion

We incubated G6PD with increasing concentrations of Gdn-HCl to induce protein unfolding. Then, we estimated the residual activity level of the enzyme, which is indicative of its folding state. Figure 4B shows a progressive loss of residual activity that is Gdn-HCl concentration-dependent, since at low concentrations of Gdn-HCl (0 to 0.5 M), enzymatic activity did not change, while at concentrations ranging from 0.6 to 0.80 M, the activity decreased gradually until it become imperceptible. The value of (Gdn-HCl)_1/2_ determined for recombinant G6PD from GDI was higher than that previously reported for the G6PDs of *Homo sapiens* and *G. lamblia* (0.3–0.45 M) [38,40,41,42] and lower with respect to G6PD from *B. malayi,* where the C_1/2_ value was 1.2 M of Gdn-HCl [25]. In addition, we evaluated the effect of Gdn-HCl on the activity of recombinant G6PD in the presence or absence of their physiological substrates (1 K_m_ for NADP^+^ = 7.2 µM and 1 K_m_ of G6P= 63 µM), and no changes were observed in the residual activity compared to those found without any substrate, as shown in Figure 4B. The resistance to structural damage of G6PD from GDI that we observed could be related to the natural habitats of the organism.

Protein digestion with trypsin is usually used to test the resistance to proteolysis; therefore, we evaluated the effect of trypsin on the activity of G6PD protein in the presence or absence of their two physiological substrates (NADP^+^ and G6P). As shown in Figure 4C, as the trypsin concentration increased (from 0 to 1 mg/mL), the same susceptibility to trypsin digestion as that of the native enzyme was observed in the presence of 1 K_m_ of both G6P and NADP^+^. Furthermore, we observed that at 0.47 mg/mL of trypsin, the G6PD protein lost 50% of its initial activity, even when the enzyme was incubated in the presence of its physiological substrates.

### 2.5. Spectroscopic Characterization

#### 2.5.1. Circular Dichroism (CD) Analysis and Thermal Stability

The secondary structure of the G6PD protein was evaluated using CD to obtain information about the proportions of α-helices and β-sheets in the native state. The G6PD protein showed the minimum absorption peaks at 210 and 222 nm in the far-UV region, which represents the content of α-helices and β-sheets without alterations (Figure 5A). On the other hand, it is interesting to note that the pattern and intensity of CD spectra for the G6PD was consistent with the α-β structure of recombinant human G6PD [35,36,40,41,42,43,44,45] and that previously reported for G6PD from *Y. lipolytica* [37]. Another approach to assessing alterations of the protein structure was thermal denaturation using CD analysis. As shown in Figure 5B, we obtained a two-state process and calculated the *T*_m_ (temperature at which half of the α-helices were unfolded) for the G6PD enzyme at 50.6 °C. Finally, it is important to mention that the *T*_m_ obtained for the G6PD was different from that previously reported in recombinant human WT G6PD, where a *T*_m_ of 59.5 °C was found [35,36,40,41,42,43,44,45,46,47,48].

#### 2.5.2. Intrinsic Fluorescence

The intrinsic fluorescence of six tryptophan residues contained in the G6PD/monomer was monitored to determine structural changes as the Gdn-HCl concentrations were increased. The intrinsic fluorescence for the native G6PD protein without Gdn-HCl showed a peak at 337 nm with a maximum intensity of 603 arbitrary units (a.u.) (Figure 6A), while for the highest concentration of Gdn-HCl (1 M), the maximum intensity of fluorescence decreased by 40% (387 a.u.) compared to without Gdn-HCl (Figure 6B). It is also interesting to note that changes in the maximum emission length (λ_max_) of G6PD caused by increasing concentrations of Gdn-HCl were observed (Figure 6C). As previously noticed, the decrease in fluorescence intensity and change in the maximum emission length (λ_max_) could be because six tryptophan residues were exposed to the environment after Gdn-HCl-induced unfolding of the native 3D structure.

### 2.6. Homology Modeling of the Recombinant *G6PD*

The predicted and annotated sequence of G6PD from GDI submitted to the UniProt database [49] and BLAST protein data bank (PDB) was performed using a target database of 3D structures. The data showed that, based on its crystallographic 3D structures, the G6PD sequence of GDI has an identity of 38.9% with the G6PD from *Mycobacterium avium* (PDB entry 4LGV), 37.1% with the G6PD of *Homo sapiens* (PDB entry 2BH9), 35% with the *Trypanosoma cruzy* (PDB entry 5AQ1)*,* and 34% with *Leuconostoc mesenteroides* (PDB entry 1H9A). Due to the existence of crystallographic 3D structures of G6PD enzymes, the sequence was submitted to the Swiss Model Server [50] (ZWF model.pdb). The entire 3D G6PD of the GDI model showed a total of 16 α-helices and 15 β-strands (Appendix A), and the N-terminal (amino acids 1–197) was found to contain the β-α-β Rossmann type folding domain, where the binding sites of β-d-glucose-6-phosphate (G6P) and NADP^+^ are located. The structural alignment of the G6PD model of GDI with the human G6PD enzyme (PDB entry 2BH9) and G6PD from *L. mesenteroides* (PDB entry 1H9A) also showed the presence of the β-α-β Rossmann type folding domain, as well as the β + α domain that forms the dimer interface and contains a large antiparallel sheet (Appendix A) [19,51]. Interestingly, although the G6PD of GDI showed the β + α domain located in the C-terminal region, we consider that this enzyme does not have the ability to bind to the second structural NADP^+^ because the thermostability data suggest that, even in the presence of NADP^+^, the protein is not protected from degradation. This is consistent with the idea that this site is only present in higher organisms where it has been proposed to be involved in the dimerization and stability of the enzyme [47,51]. Finally, we also observed that the G6PD sequence of GDI has three conserved fragments that are present in other G6PDs from different organisms. Appendix A shows these fragments as follows: RIDHYLGKE, which is involved in substrate binding and catalysis through lysine (Lys204), GxGGDLT, which is involved in the binding of the catalytic NADP^+^ coenzyme, and EKPxG, which contains the amino acid proline (Pro 171), which is related to the correct positioning of the substrate (G6P) and coenzyme (NADP^+^) during the enzymatic reaction (Appendix A). The presence of these motifs corroborates the functionality of the enzyme.

## 3. Materials and Methods

### 3.1. Cloning of zwf from GDI

The *zwf* gene from Reference Sequence GenBank: CAP54231.1 was amplified by polymerase chain reaction (PCR) using template genomic DNA from GDI and specific primers. The forward primer 5′-TAATCATATGGCTCAGCTCCCCCC-3′ and reverse primer 5′-ATTAGGATCCTTACGCCATGTCCTCGT-3′ contained *Nde*I and *Bam*HI restriction sites (underlined), respectively. The reaction mixture consisted of a 200 ng primer, 10 mM dNTP mixture, 1× PCR buffer, and 1 U of phusion^®^ high fidelity DNA polymerase (Thermo scientific, Hudson, NH, USA). The PCR conditions used for the amplification of the *zwf* gene were as follows: 2 min at 98 °C for denaturation, 25 cycles of amplification (35 s at 98 °C, 20 s at 65 °C, 30 s at 72 °C), and 5 min at 72 °C for extension. The PCR product was separated using 1% agarose gel electrophoresis, stained with GelRed (Nucleic Acid Gel, Biotium, Fremon, CA, USA), and visualized in a MultiDoc-It Digital Imaging System (UVP, Upland, CA, USA). The amplicon of the expected size (1530 bp) was purified and ligated into the pJET 1.2 vector (CloneJET PCR Cloning Kit; Thermo Scientific, Hudson, NH, USA), following the instructions of the protocol, to yield the plasmid named pJET/*zwf*, which was used to transform *E. coli* TOP10F’ competent cells. Then, from the transformant colonies, the plasmid DNA was extracted using the GeneJET Plasmid Miniprep Kit (Thermo Scientific, Waltham, MA, USA), according to the manufacturer’s instructions, and bidirectional DNA sequencing with verified internal forward and reverse sequencing primers was used to confirm that desired recombinant plasmids were obtained. The verified sequence was digested with the restriction enzymes *Nde*I and *Bam*HI and sub-cloned into the pET3a-HisTEVP plasmid to generate the pET3a-HisTEVP-*zwf* and used to transform *E. coli* BL21(DE3)pLysS competent cells (Invitrogen, Carlsbad, CA, USA). The transformant was selected from the Luria Bertani (LB) plate medium containing 100 μg/mL ampicillin.

### 3.2. Alignment of the *G6PD* Protein of GDI

A total of 60 amino acid sequences were retrieved from Swiss-Prot and NCBI (Appendix A). From Swiss Prot, 48 sequences were selected, as they are manually curated, and to avoiding having more than one representative from a species. A total of 12 additional species were selected from NCBI [13], as they were related to the clade of interest and have no representatives in the Swiss Prot database [49]. Multiple sequence alignment was performed with MAFFT V7.427 [15] using the “auto” option to select the most appropriate alignment strategy according to the dataset, while the rest of the parameters were left as default. The alignment was visualized with the online program Jalview [15]. A maximum likelihood phylogenetic reconstruction was performed with FastTree 2.1.10 (Berkeley, CA, USA) [52] using the Whelan and Goldman model and 1,000 bootstraps. The evolutionary distances were computed using the Poisson correction method and are represented in units of the number of amino acid substitutions per site. The analysis involved 60 amino acid sequences. All positions containing gaps and missing data were eliminated. There was a total of 325 positions in the final dataset.

### 3.3. Expression and Purification of the *G6PD* Protein

The *E. coli* BL21(DE3)pLysS cells containing the recombinant pET3a-HisTEVP-*zwf* plasmid were used to inoculate 50 mL of fresh LB medium. This medium was then transferred into 2 L of fresh LB medium with 100 μg/mL ampicillin at 37 °C for 5 h at 180 rpm. When the cell density reached an optical density (OD_600_) of 0.6 to 0.8, the protein was induced by adding 0.3 mM IPTG at 30 °C for 12 h. The cells were collected, harvested, suspended in a lysis buffer (0.1 M Tris-HCl, pH 7.6, 0.1% β-mercaptoethanol, 0.5 mM PMSF, 3 mM MgCl_2_, and 10% of glycerol), and lysated by sonication. The crude extract was obtained by centrifugation at 18,000× *g* for 25 min at 10 °C and applied to a Ni Sepharose high performance column (GE Healthcare, Chicago, IL, USA) that was pre-equilibrated with a binding buffer (0.1 M Tris-HCl, 50 mM NaCl, pH 8.0 and 10% of glycerol). Then, the column was washed (binding buffer plus 50 mM imidazole), and the protein was eluted with the same binding buffer plus 250 mM imidazole [35]. Imidazole was removed from the sample by five consecutive dilutions and concentrated using a microcon-10 kDa centrifugal filter unit (Millipore). The G6PD protein was digested with the TEVP protease (previously purified) to remove additional amino acid residues corresponding to the (His)6-tag sequence, and site-specific protease HisTEVP recognition was added in the N-terminal region of the G6PD protein. The TEVP protein used for the digestions was purified by a Ni Sepharose high performance column (GE Healthcare, Chicago, IL, USA) [35]. Finally, the purified G6PD protein was analyzed in 12% SDS-PAGE gel and stained with colloidal coomassie brilliant blue (R-250) (Sigma-Aldrich, San Luis, Misuri, USA). The protein concentration was quantified in accordance with Lowry et al. [53] using bovine serum albumin as the standard. To preserve the protein, glycerol was added to the purified protein (50 % *v*/*v*) and stored at −70 °C.

### 3.4. Biochemical Analysis of *G6PD* Protein from GDI

#### 3.4.1. Native Status of the G6PD protein

The native state of the purified G6PD protein was determined using gel filtration chromatography (GFC). The G6PD protein was loaded into a sephacryl 200 (16/60) gel filtration column (GE Healthcare, UK) pre-equilibrated with equilibrium buffer T (100 mM Tris-HCl buffer pH 7.35, 0.01 M MgCl_2_), and the protein was eluted using the equilibrium buffer at a flow rate of 0.3 mL/min, with the absorbance signal monitored at 280 nm. Using the same conditions, the column calibration was done by employing gel filtration standard # 151-1901 (Mark Bio-Rad). In both cases, the column was coupled to the AKTA pure fast protein liquid chromatography (FPLC) system (GE Healthcare, Chicago, IL, USA).

#### 3.4.2. Effect of pH and Temperature on G6PD Activity

To evaluate the effect of pH on the activity of G6PD from GDI, we measured the activity at a pH range from 2.6 to 10.0 using four different buffer systems: Mcllvaine buffer (pH 3.0–6.0), 50 mM MES buffer (pH 6.0–6.75), 50 mM HEPES buffer (pH 6.75–8.0), 50 mM Tris buffer (pH 8.0–9.0), and glycine (pH 9.0–10). G6PD activity was determined as previously reported by Gómez-Manzo et al. [35,36]. The non-enzymatic reduction of NADP^+^ in the assay was measured at each pH value and subtracted from the experimental points. Moreover, we evaluated the effect of temperature on G6PD activity. The enzyme was incubated at a final concentration of 0.2 mg/mL at different temperatures from 15 to 60 °C for 20 min according to a previously reported method [36], and then the activity was determined and expressed as a percentage. The enzyme activity incubated at 37 °C was set to 100%. All thermal inactivation trials were performed in triplicate.

#### 3.4.3. Kinetic Study of the G6PD Enzyme

The experimental steady-state kinetic parameters for the G6P substrate were obtained from the initial velocity data by varying the G6P substrate (0 to 2 mM), while the NADP^+^ was fixed at a saturating concentration (∼10-fold of the K_m_ value; 72 µM). To obtain the steady-state kinetic parameters for the NADP^+^ substrate, this value was varied from 0 to 250 µM, and the G6P substrate was fixed at a saturating concentration (∼10-fold of the K_m_ value; 630 µM). The initial velocity obtained for each concentration was used to calculate the rate of product formation of the NADPH (µmol/min/mg) using the extinction coefficient of reduced NADPH at 340 nm (6220 M^−1^·cm^−1^). The K_m_, *k*_cat_, and *V*_max_ parameters were obtained by fitting the data to the Michaelis–Menten equation by non-linear regression calculations [36]. All the trials were performed in triplicate.

### 3.5. Evaluation of Protein Stability

#### 3.5.1. Thermal Inactivation Analysis

The effect of temperature on the stability of the protein in the presence of its physiological substrates (G6P and NADP) was evaluated by thermal analysis. The G6PD protein was suspended in a T buffer at a final protein concentration of 0.2 mg/mL and incubated with 1 K_m_ of NADP^+^ (7.2 µM) and G6P (63 µM) for 20 min at different temperatures ranging from 37 to 60 °C, as previously reported [35,36]. Later, the residual activity was measured spectrophotometrically. The residual activity of the enzyme incubated at 37 °C was fixed at 100%. All thermal inactivation trials were performed in triplicate.

#### 3.5.2. Stability of G6PD in the Presence of Gdn-HCl

Another way of assessing protein stability was in the presence of Gdn-HCl, which was incubated with 1 K_m_ of NADP^+^ (7.2 µM) or G6P (63 µM). The G6PD protein was adjusted to 0.2 mg/mL in a T buffer and incubated at 37 °C for 2 h under different concentrations of Gdn-HCl (from 0 to 1 M) in the presence or absence of 1 K_m_ of NADP^+^ (7.2 µM) or G6P (63 µM). Then, the activity of G6PD was determined, and the residual activity of the the enzyme incubated at 37 °C in the absence of Gdn-HCl was fixed at 100%. The experiment was performed in triplicate.

#### 3.5.3. Stability of G6PD in the Presence of Protease Digestion

Finally, the stability of the G6PD protein was evaluated by its susceptibility to protease digestion with trypsin in both the absence and presence of 1 K_m_ of NADP^+^ (7.2 µM) or G6P (63 µM). The protein was adjusted to 0.2 mg/mL in T buffer and incubated with different trypsin concentrations ranging from 0 to 1 mg/mL at 37 °C for 3 h, and before the residual activity was measured, the reaction was stopped with 5 mM of PMSF [42]. The residual activity of the enzyme incubated without trypsin was fixed at 100%. The experiment was performed in triplicate.

### 3.6. Homology Modeling of the Structure of *G6PD* Protein

The homology model of the full-length G6PD protein from GDI was built on the Swiss-Model server [50] using the crystal structure of G6PD from *L. mesenteroides* (PDB entry 1H9A; [27]) as a template. The best model was selected according to the GMQE, QSQE, and QMEAN statistical parameters. Then, the generated model was subjected to energy minimization using the YASARA software [54] and validated using MolProbity [55]. Structural analysis was performed by manual inspection using Coot [56]. The graphical representations were prepared with the CCP4mg software [57].

### 3.7. Spectroscopic Characterization

#### 3.7.1. Circular Dichroism (CD) and Thermal Stability

The secondary structure of the G6PD protein was analyzed by CD, as previously described [35]. Purified protein was adjusted at 0.5 mg/mL in 50 mM phosphate buffer at pH 7.4 and loaded in a rectangular quartz cuvette with a 1 cm optical path. Spectra were recorded at 25 °C in the ultra violet circular dichroism (UV-CD), ranging from 200 to 260 nm with a scanning speed of 50 nm.min^−1^. Spectra of the blank were subtracted from all the obtained spectra that contained the protein. The curves were plotted using the Origin program.

Furthermore, thermal denaturation of the G6PD was monitored at 222 nm as a function of temperature, ranging from 20 to 90 °C, and increasing at a rate of 1 °C/2.5 min. The protein concentration was adjusted to 0.5 mg/mL in 50 mM of phosphate buffer at pH 7.4. The assay was performed in a spectropolarimeter (Jasco J-8190 ^®^, Easton, MD, USA), as previously reported [43]. The data were adjusted with the Boltzmann equation using the Origin version 8 (2016) program. Both experiments were performed in duplicate at 25 °C.

#### 3.7.2. Intrinsic Fluorescence of the G6PD Protein

The intrinsic fluorescence of the six tryptophans present in the G6PD protein was determined in the presence of Gdn-HCl. The protein was adjusted at 0.1 mg/mL in 50 mM of phosphate buffer at pH 7.4 and incubated at 37 °C for 2 h in the presence of Gdn-HCl (from 0 to 1 M). the spectra were recorded at 310–500 nm in a Perkin-Elmer LS-55 fluorescence spectrometer (Perkin Elmer, Wellesley, MA, USA) using an excitation wavelength of 295 nm and slits of excitation and emission of 4.5 and 3.7 nm, respectively. The final spectrum was the average of five scans; later, each spectrum was subtracted from the spectra of the blank (without protein).

## 4. Conclusions

We reported, for the first time, the cloning, purification, and characterization of G6PD from the genome of *Glunacetobacter diazotrophicus,* PAL5. The purified protein allowed us to determine different structural and functional parameters. In the kinetic analysis, the enzyme showed specificity for the G6P and NADP^+^ substrates, which allowed us to determine the values of *Km* and *Vmax*. However, when this protein was incubated with NADP^+^, no protective effect on the thermal stability was observed, in contrast to other previously reported G6PDs. Bioinformatics analyses and the 3D model revealed that the protein contains highly conserved fragments that participate in catalysis and substrate binding, similar to eukaryotic organisms. Finally, this new data allowed us to obtain a better understanding of the metabolism of this endophytic bacterium and offered a better idea of its potential for use in biotechnology applications due to the production of metabolites of biotechnological interest by this bacterium.

## Figures and Tables

**Figure 1 ijms-20-05279-f001:**
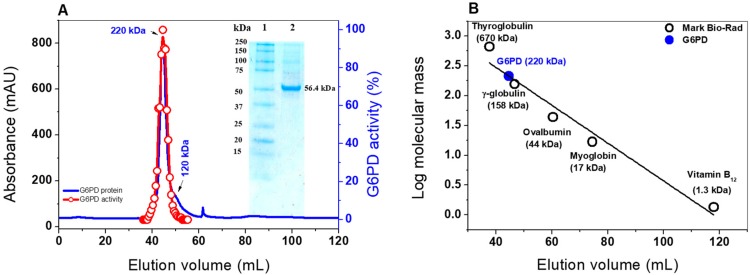
Oligomeric status and SDS-PAGE of purified G6PD protein from GDI. (**A**) FPLC chromatograms of the purified G6PD protein. The blue line represents the protein absorptivity at A280 nm. The red line represents the G6PD activity. Inset: SDS-PAGE analysis of the expressed G6PD protein; Lane 1: protein MW marker precision plus protein kaleidoscope standards from Bio-Rad; Lane 2. purified G6PD protein. Ten micrograms of protein were loaded, and SDS-PAGE was stained with colloidal Coomassie Brilliant Blue (R-250) (Sigma-Aldrich). (**B**) Calibration curve showing the elution volumes versus the log of MW Bio-Rad’s gel filtration standard (black spots). The MW of G6PD is shown on the straight line obtained (blue spot).

**Figure 2 ijms-20-05279-f002:**
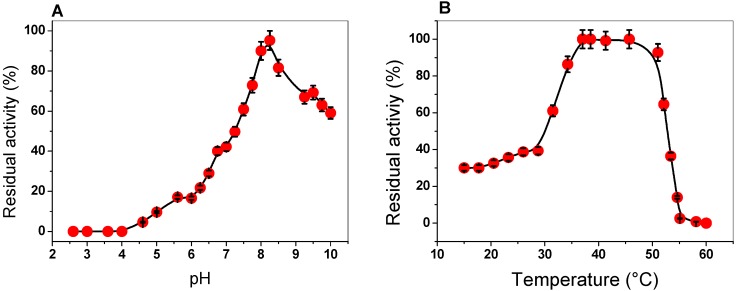
Effect of pH and temperature on the activity of the G6PD enzyme. (**A**) Effect of pH on G6PD activity. (**B**) Heat-inactivation profile of G6PD activity. Error bars indicate the mean ± standard deviation of the triplicate values.

**Figure 3 ijms-20-05279-f003:**
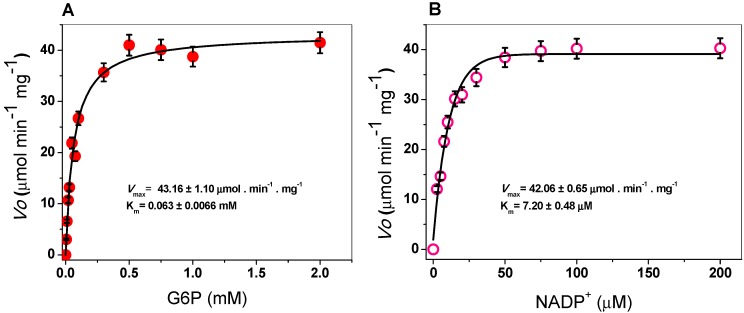
Michaelis–Menten plots for G6PD from *Gluconacetobacter diazotrophicus* (GDI) with (**A**) G6P and (**B**) NADP^+^ as substrates. The data represent the mean ± SD from five independent experiments.

**Figure 4 ijms-20-05279-f004:**
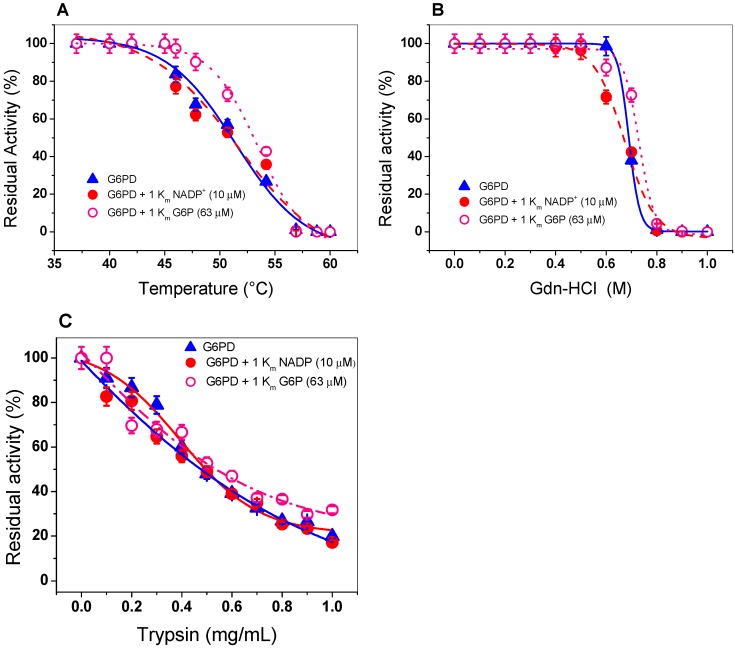
Evaluation of protein stability. The G6PD enzyme was incubated in the (Δ) absence or presence of (•) NADP^+^ (1 Km = 7.2 µM) and (o) G6P (1 Km = 63 µM). (**A**) Thermal inactivation assays of G6PD protein. (**B**) Stability of G6PD protein in the presence or absence of Gdn-HCl. (**C**) Stability of G6PD in the presence or absence of protease digestion. In all cases, the G6PD protein was incubated at 0.2 mg/mL, and the residual activity was measured with 200 ng of total protein. All the assays were performed in triplicate; standard errors were lower than 5%.

**Figure 5 ijms-20-05279-f005:**
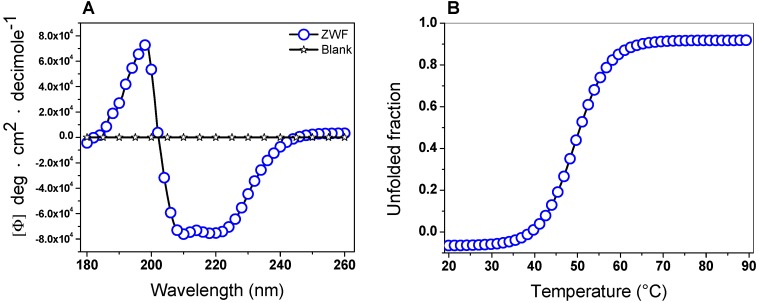
Circular dichroism (CD) analysis and thermal stability. (**A**) Far-ultraviolet (UV) CD spectra and (**B**) thermal stability of the G6PD protein. Changes in the CD signal were monitored at 222 nm as the temperature increased (20 to 90 °C). In both assays, the G6PD protein was recorded at 0.5 mg/mL in a 25 mM phosphate buffer (pH 7.4). This experiment is representative of duplicate experiments.

**Figure 6 ijms-20-05279-f006:**
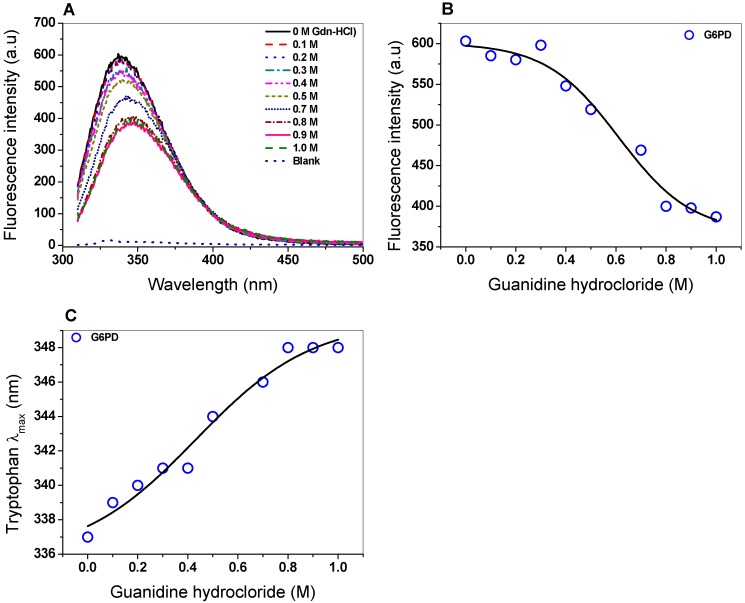
Spectroscopic characterization. (**A**) Intrinsic fluorescence spectra of the G6PD protein in the presence of Gdn-HCl; (**B**) fluorescence intensity obtained with different concentrations of Gdn-HCl. (**C**) Maximum emission intrinsic fluorescence of tryptophan produced by denaturing induced by Gnd-HCl. G6PD (0.2 mg/mL) was incubated in PBS (50 mM, pH 7.4) in the presence of Gdn-HCl. The assays were realized by triplicate (standard errors < 5%).

**Table 1 ijms-20-05279-t001:** Steady-state kinetic parameters of the previously reported G6PDs.

Organism	k_cat_ (s^−1^)	K_m_ G6P (µM)	K_m_ NADP^+^ (µM)	Reference
*Gluconacetobacter diazotrophichus*	293,181	63	7	This study
*Escherichia coli* DH5α	32	224	127	[32]
*Pseudomonas aeruginosa*	540	498	56	[21]
*Termotoga maritima*	35,000	200	40	[20]
*Haloferax volcanii*	11	370	520	[26]
*Giardia lamblia*	31	18	14	[38]
*Plasmodium falciparum*	8	19	6	[39]
*Trypanosoma cruzy*	62	77	16	[31]
*Aspergillus niger*	NR	153	26	[34]
*Aspergillus oryzae*	1000	109	6	[33]
*Brugia malayi*	40	245	14	[25]
Dog liver	NR	122	10	[29]
Buffalo liver	NR	NR	59	[24]
Camel liver	NR	81	81	[22]
*Homo sapiens*	230	38	7	[36]

NR = Data not reported.

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
