# Peer review of "Molecular Cloning and Exploration of the Biochemical and Functional Analysis of Recombinant Glucose-6-Phosphate Dehydrogenase from Gluconoacetobacter diazotrophicus PAL5"

_ijms, 2019, doi:10.3390/ijms20215279_

Round 1
Reviewer 1 Report
Detailed characterisation of novel enzymes is highly welcome. Nevertheless, the Authors must address some questions.
1) General comments
Extensive editing of English language and style is required. There are the grammar errors (e.g., p.2 lane 15). The typos and the punctuation are very poor (e.g., the sentence p.2 lines 32-34). Frequently, the Authors use the jargon or mental shortcuts (e.g., "evolutionary story"). The structure of the text should be improved. Some parts need to be rewritten (e.g., the whole legend for figure 1, p11 lines 5-15 - repetition).
The mutagenesis of the crucial residues (e.g. some of the amino acids from mentioned motifs) should abolish enzymatic activity. Such an experiment would strengthen the work significantly.
2) Specific comments for revision:
a) major:
the alignment(s), the tree(s), and the proteins model(s) should be added as supplementary material or stored in the external databases (e.g., FigShare). Without them it is impossible to validate the correctness of the bioinformatic results why only 60 proteins had been used for the alignment? How they have been selected? the whole figure 1 should be made again (numbering should be after GDI sequence, GDI sequence should be at the top, the colors should represent the conservation, the three mentioned motifs should be highlighted) the text explaining the figure 1 should be also rewritten (why "Proline 171" is as upper case letter? Why the numbering in the text is according GDI while in fig. 1 according L. mesentroides?, etc.) when using some bioinformatic methods always specify the parameters (e.g. was MAFFT was used with default parameters?, how did you predicted isoelectric point?) - this is not explained neither in the results nor in the methods the figure 2 lack bootstrap values (this is crucial to asses tree quality), the GDI branch could be "bolded" the activity of G6PD in various pH and temperature should be checked in wider ranges (pH below 6 and temperature below 37oC) - the bacteria is known to be acid-tolerant and lives in the plans replace BLAST with HHblits or any other profile-profile method for homology detection from the text and the figure 9 legend it is unknown what protein structure had been used as the template (hint: prefer the tetrameric structures if possible) redo the fig 9 and legend (e.g. 9a contains green helices not midnight blue, mark three motifs in some color) add more information about the programs parameters in the sections "3.2. Alignment of the G6PD protein of GDI" and "3.6. Homology modeling of the structure of G6PD protein"
b) minor:
remove star (*) for Jaime Marcial-Quino who is not a corresponding author what is "evolutionary story"? there is two Figures 6 while no Figure 5 rewrite "Because there is no information on the secondary structure of the G6PD protein," as it is not true because there are structures in PDB if you use bioinformatics tools cite them (lack of citation for BLAST, UNIPROT) I do not know UniProt program, I only know UniProt database rewrite "Furthermore, we performed an alignment with the crystal structure of G6PD previously reported in Homo sapiens (PBD 2BH9)" - how the alignment has been done? What tool has been used?
Reviewer 2 Report
Review of the manuscript entitled: “Molecular Cloning and exploration of the biochemical and functional analysis of recombinant Glucose-6-phosphate dehydrogenase from Gluconoacetobacter diazotrophicus PAL5” submitted to the International Journal of Molecular Sciences.
In the article, the authors document several parameters related to the activity, stability and putative 3D conformation of the G6PD enzyme expressed by Gluconoacetobacter diazotrophicus. The scientific strategies are adequate and the results are clearly presented in the different figures. The reported data could be of interest to biochemists working on G6PDH and to researchers interested in Gluconoacetobacter diazotrophicus as a model organism or biotechnological tool.
However, there are several problems in the main text: imprecision, incomplete explanations, grammatical errors, error in figure numbering, repetitions, incorrect or oversimplified statements, etc.
The list below details the main points that should be addressed:
- according to Uniprot, the gene encoding G6PD in Gluconoacetobacter diazotrophicus is zwf. This might be worth documenting in the article (result section, legend of figure 1, material and methods section,…).
- It should be mentioned, in the result section, how the enzyme was produced and purified, if the tag is in N- or C-ter position, and whether it was removed after purification of the protein.
- p2, line 42: this is the first time that the sequence was cloned by the authors. I would suggest using the verb “analyze” instead of “confirm” in the first sentence of the result section.
-p2, line 45-47. Was it a protein blast or cDNA blast analysis?
- P.4, line 13: I would precise in the text that the size of the protein (monomer) was estimated after SDS-PAGE.
- p4, line 12-16: What is the reason for the differences of molecular mass detected between the GDI-derived G6PD protein and mammalian G6PD forms?
-p4, line 20: At this point in the manuscript, the authors have not proven that the tetramer is active. The detection of the protein in the elution fractions was based on absorbance values. Therefore, I recommend removing the word “active” from the sentence starting on line 19.
- “Molecular weight” and “Molecular mass” are used interchangeably in the manuscript when referring to the size of G6PD, or when referring to the size of marker proteins. Please clarify.
- p4, line 22: the authors point out that G6PD could form a dimer of approx. 120 kDa (second peak after FLPC). The next sentence on page 5 states that there are no homodimers found after FPLC… Please clarify.
-p5, line 4 : what do you mean by switch between dimer and tetramer?
-p5, line 4: as I understand it, the point of this sentence is to document that G6PD tetramers are also found in other species. Why are rat liver G6PD hexamers listed in this sentence?
-p5, paragraph 2.3.1. Additional information regarding the type of tests that were conducted would be welcome. How was the activity measured? While it is explained in the material and methods section, the story would be easier to grasp if a few pieces of information were added in the main text. This comment also applies to other parts of the result section.
-p5, line 22: what is an “early stationary phase”?
-p5, line 23: NAPDH has other roles, besides its involvement in lipid synthesis
- p5, lines 29-30: The wording may raise the following question. Was there a continuous rise in temperature, or were different samples incubated at different temperatures?
-p6, line 7: It is stated that the second substrate is added at a concentration of approx. 10 fold the Km value, which would mean 630 µM for G6P and 72 µM for NADP+. In the material and method section (p13), 1 mM is listed for G6P and 250 µM for NADP+. Please modify the section that is incorrect.
- p6, Figure 6: I think that this is Figure 5 and not Figure 6.
- p6, Figure 6 (actually 5): please check SD values associated with the Km constant on the left graph. There are two SD values.
-p7, line 6: the Km value of the reaction catalyzed by G6PD (NADP+ transformation) is reported at 10 µM, whereas it is 7.2 µM on the previous page.
-p7, line 8: was the test conducted at 50°C as stated in the text? or did the authors check the stability of the enzyme at different temperatures?
-p7, chapter 2.4: additional information on how NADP+ interacts with the human enzyme and how it can stabilize this enzyme would be helpful. Also, regarding the conclusion of this paragraph, I am not convinced that the authors have sufficient data to conclude that the G6PD enzyme from GDI lacks the amino acids required to form a structural NADP+ binding site. This point is worth discussing, but should be extended by additional pieces of information (conservation of residues in this region, comparison with other G6PD enzymes from other species, hypotheses based on the 3D prediction model, … ).
-p7, line 10. What is the reference of the article in which the T50 of human G6PD was reported? As a point of comparison, it might be worth mentioning that the T50 of the human enzyme raises from approx. 47°C to approx. 59°C when in the presence of NADP+ (data from : Catalysts 2017, 7(5), 135; https://doi.org/10.3390/catal7050135).
-p7, line 21: Gdn-HCl should be defined as guanidine hydrochloride at least once
-p7, line 23-24: “unfolding profile pattern” and “unfolding curve” are ambiguous terms when used at the beginning of this paragraph, considering that the test does not actually analyze the conformation of the protein. I would suggest stating first that the test is a measure of the enzyme activity level after incubation with different concentrations of Gdn-HCl. Then, the point could be made that the residual activity level is indicative of the folding state of the enzyme.
-p7, line 35: I suggest writing that trypsin is used to test the resistance to proteolysis.
- p8, line 14: do the authors mean secondary structures (plural)? or tertiary structure? Or quaternary (cf. tetramers)?
- p9, Figure 7. Were standard deviations calculated from only two values as indicated in the legend? Please clarify.
-p9, paragraph 2.5.2. This paragraph is very difficult to understand. It is written that the fluorescence of the control protein is decreased? Compared to what? Also, the following sentence is confusing “the maximum intensity decreased by 40% until reaching a maximum fluorescence intensity of 387 A.U.” Please clarify.
- p11: please check for redundancy between line 3 and line 15.
- p12, line 29: the term “inoculate” refers to the entry of a microorganism in a medium, not to the entry of a plasmid within a microorganism.
- p14, line 36. What is the basis of the statement “the enzyme showed high specificity for the G6P and NADP+ substrates”? Were other substrates tested?
- p14, in the conclusion. The study would be strengthened by a longer explanation of what are the putative biotechnological applications in which GDI could be used.
- A general comment: there are several shortcuts when referring to the enzyme throughout the article. Some examples:
- p.2, line 45: “…. found 100% similarity with the sequence of GDI” . The authors are comparing G6PD sequences with one another; this is not a comparison with the GDI organism as a whole.
p.3, line 6: “… corresponding to the GDI sequence”. Do the authors mean “corresponding to the G6PD sequence encoded by the GDI genome”?
-p.4, line 12-16: the structure of the sentence indicates that the listed species and organs do have a MW…
- Same type of shortcut in the legend of Fig.2.
- p14, in the conclusion, line 39: the sentence suggest that the protein is a prokaryotic organism…
Etc.
Lastly, I detected some grammatical errors, as well as sentences structure problems. I only listed some of them below. I would recommend asking a native English speaker to revise the manuscript.
- p2, line 15: performs instead of perform
- p2, line 32: the following sentence is very long and its meaning is unclear. “Given the relevance ……….G6PD from GDI”.
- p3, line 11-12 : binding of the substrate to human G6PD
- p3, line 12 : please check the use of the term “catalysis” in the last sentence of the paragraph. An enzyme catalyzes a reaction, not a substrate.
- p4, line 1 : “Phylogenetic analysis revealed that three clades were descended from? the G6PD sequence found in GDI”?
- p.5, line 3 :” a tetrameric structures (singular)”
-p5, line 19: I would replace “pH levels” by “different pHs” or by “different pH values”
-p5, line 25: please refer to the enzymes and not to the organisms when discussing optimal pH values.
-p5, line 29: “To investigate”? instead of “To determinate”?
-p7, line 30: revise sentence starting by “In addition….”
-p8, legend of figure 6: there is a “l” missing at the end of Gdn-HC…
Round 2
Reviewer 1 Report
1) General comment
The authors did not make the requested improvements/corrections, thus I recommend to reject the article. In general, the English language and the structure have been improved, but in many places, the manuscript is still hard to read.
2)Specific comments NOT addressed by the authors:
a) major:
- the alignment(s), the tree(s), and the protein model(s) were not added to the supplementary material or stored in the external databases (e.g., FigShare). Without them, it is impossible to validate the correctness of the bioinformatic results presented in the manuscript. This point alone is sufficient to retract the manuscript.
- figure 2 lacks bootstrap values (this is crucial to asses tree quality).
- the star (*) for Jaime Marcial-Quino who is not a corresponding author has not been corrected (this may be considered a minor flaw, but it is actually not, if you consider how the science funding is distributed), this was explicitly stated to fix.
b) minor
- the Authors did not replace BLAST with HHblits or any other profile-profile method for homology detection (such methods are much more accurate).
- the Authors did not rewrite "Furthermore, we performed an alignment with the crystal structure of G6PD previously reported in Homo sapiens (PBD 2BH9)" - It is unknown how the alignment has been done (was that structural or sequence alignment). What tool has been used?
Author Response
October 16th, 2019.
Dear reviewer #1:
We are very grateful for your valuables observations performed that improved this manuscript
Comment #1: - 1) General comment
The authors did not make the requested improvements/corrections, thus I recommend to reject the article. In general, the English language and the structure have been improved, but in many places, the manuscript is still hard to read.
Response 1:
We have read the manuscript and rewrite the sections that we consider are hard to read. We consider that has been improved the manuscript and we hope to have done all the observations performed for you.
2)Specific comments NOT addressed by the authors:
a) major:
Comment #2: - The alignment(s), the tree(s), and the protein model(s) were not added to the supplementary material or stored in the external databases (e.g., FigShare). Without them, it is impossible to validate the correctness of the bioinformatic results presented in the manuscript. This point alone is sufficient to retract the manuscript.
Response 2:
The alignment(s), the tree(s), and the protein model(s) were added to the supplementary material.
Comment #3: - Figure 2 lacks bootstrap values (this is crucial to asses tree quality).
Response 3:
In figure legends (Figure 2) we add the bootstrap values which was of 1000.
This information has also been added in the Materials and Methods section as follow: “The phylogenetic tree was constructed in MEGA7 (v. 7.0.21) [51], which was inferred using the Neighbor-Joining method and 1000 bootstraps.”
Comment #4: - The star (*) for Jaime Marcial-Quino who is not a corresponding author has not been corrected (this may be considered a minor flaw, but it is actually not, if you consider how the science funding is distributed), this was explicitly stated to fix.
Response 4:
The star (*) for Jaime Marcial-Quino was removed and corrected in the manuscript.
b) minor
Comment #5: - The Authors did not replace BLAST with HHblits or any other profile-profile method for homology detection (such methods are much more accurate).
Response 5:
We replaced BLAST by HHblits and the Table S1 was modified.
Furthermore, we added the reference [14] in the manuscript:
[14] Remmert, M.; Biegert, A.; Hauser, A.; Söding, J. HHblits: lightning-fast iterative protein sequence searching by HMM-HMM alignment. Nature Methods, (2012), 9, 173-175.
Comment #6: - The Authors did not rewrite "Furthermore, we performed an alignment with the crystal structure of G6PD previously reported in Homo sapiens (PBD 2BH9)" - It is unknown how the alignment has been done (was that structural or sequence alignment). What tool has been used?
Response 6:
The sentence was modified according with your observation, we replaced the sentence by “The structural alignment of the G6PD model of GDI with the human G6PD enzyme (PDB entry 2BH9) and G6PD from L. mesenteroides (PDB entry 1H9A) also showed the presence of the b-a-b Rossmann type folding domain, as well as the b + a domain which forms the dimer interface and contains a large antiparallel sheet (Figure S3 A,B)”.
Furthermore, the “Structural alignment in Figure S3 B was performed using Coot [55].” This information was added in the Figure legends.
[55] Emsley, P.; Lohkamp, B.; Scott, W.G.; Cowtan, K. Features and development of Coot. Acta Crystallogr. D Biol. Crystallogr. 2010, 66, 486–501.
Finally, we added as Supplementary File the homology model for the G6PD from GDI.
Thank you very much for your observations.
Sincerely,
Dr. Saúl Gómez-Manzo
Laboratorio de Bioquímica Genética.
Torre de Investigación, 6º piso.
Av. Insurgentes sur 3700-C, México DF 04530.
Phone + (52 55) 10840900 ext. 1442
E-mail: saulmanzo@ciencias.unam.mx
Reviewer 2 Report
The authors have significantly improved the quality of their manuscript. Typos and errors were corrected and precisions were added where needed. Just a few more points:
p.3. line 20: there is a “s” missing at the end of contains
p.4. legend of Fig1: G6PD should not be in italic. Same remark for Fig2.
p.4, line 11: This sentence is still unclear. “Phylogenetic analysis revealed the presence of three clades were G6PD sequence from GDI was clustered with the most closely related organisms such as Gluconacetobacter liquefaciens and Gluconacetobacter etanii (Figure 2)”.
Do the authors mean “where” instead of “were”? Could the sentence be simplified? “Phylogenetic analysis revealed that the G6PD sequence from GDI is most closely related to the G6PD sequences found in organisms such as Gluconacetobacter liquefaciens and Gluconacetobacter etanii (Figure 2).”
p.6, line 8. I would simplify as followed: “The G6PD protein, which is fused to an N-terminal His tag, was purified by Ni Sepharose High Performance affinity column (GE Healthcare). Then, the tag was removed using the site-specific protease HisTEVP”.
p.6, line 11: very long sentence. “The inset in figure 3 shows a single band (monomer) of the purified recombinant G6PD protein with an apparent MW of 56.4 kDa that was estimated by SDS-PAGE, which corresponds to the theoretical values previously reported and that is similar with those reported for other prokaryotes such as Thermotoga maritima and Pseudomonas aeruginosa [186,197], whereas it is different for the organs of mammals, such as the camel liver, rat kidney, and buffalo since they have MWs of around 60 to 75 kDa [2018-220].”
My sugegstion: “The inset in figure 3 shows that, after SDS-PAGE, the protein is detected as a single band with an apparent MW of 56.4 kDa. This is consistent with the theoretical MW of the monomeric form of G6PD. Of note, this size is similar to those reported for G6PD proteins in other prokaryotes such as Thermotoga maritima and Pseudomonas aeruginosa [186,197]. In mammals, such as the camel liver, rat kidney, and buffalo, MWs around 60 to 75 kDa have been documented [2018-220].”
p.6, line 22: “In addition, we found a second small peak with a relative MW of around 120 kDa, which could be a dimeric form, but no G6PD activity was observed for this form (Figure 3B). Furthermore, no monomer MW or larger aggregates species were observed in the Fast Protein Liquid Chromatography (FPLC) chromatogram.”
p.6, line 28: reference 24 refers to the rat protein, which has been removed from the sentence.
p.7. line 6: “As can be observed at pH values ranging from 2.6 to 4.0, no G6PD activity was detected, while that at a pH value of 5.0, G6PD activity was found around 10%.”
p.7, line 10 “such as NADPH (notably required for lipid biosynthesis and for conferring protection against…)”
p.7, line 17: “ To investigate the effect of temperature on the activity of G6PD protein, aliquots of the enzyme were incubated at different temperatures (ranging from 15 to 60 °C),…”
p.7, line 19: please check the temperature values and sentence structure. Based on the graph, I would suggest writing: “the activity remained intact from 37 to 50°C, slowly decreased between 47 and 52°C, then sharply dropped. Furthermore, ……………….when incubated at temperatures below 30°C”
p.8, line 13: “As shown in Figure 6A, when the enzyme was incubated with 1 Km (7.2 μM) of NADP+, no protective effect on thermal stability was observed, since its temperature T50 (temperature at which the enzyme loses 50%) remained similar with or without NADP+. This result is in disagreement with that previously reported for recombinant human G6PD, whose T50 increases from 47°C to 59°C when in the presence of NADP+ (46).”
p.9, line 7: “that has been reported to participate…”
p.9, line 18. I would remove the first sentence and start directly by “We incubated G6PD with increasing concentrations of Gdn-HCl to induce protein unfolding. Then, we estimated the residual activity level of the enzyme, which is indicative of its folding state. Figure 6B shows a progressive loss of residual activity that is Gdn-HCl concentration-dependent, since…”.
p.9, line 28 : “In addition, we also evaluated…”
p9, line 30 “no changes were observed …”
p.11, lines 17-22; The part starting by “As previously noticed…” is very difficult to follow. Please revise the structure of this paragraph. I suppose that the authors mean that tryptophan residues could be exposed to the environment after Gdn-HCl-induced unfolding, but the structure of the sentence is confusing. The point regarding exposure to high Gdn concentrations also needs to be clarified (concentrations >0.6 M) did have a high effect on the protein…
Author Response
October 16th, 2019.
Dear reviewer #2:
We are very grateful for your valuables observations performed that improved this manuscript
Comments and Suggestions for Authors
The authors have significantly improved the quality of their manuscript. Typos and errors were corrected and precisions were added where needed. Just a few more points:
Comment #1: - p.3. line 20: there is a “s” missing at the end of contains
Response 1:
Yes, we agree. The sentence was modified.
Comment #2: - p.4. legend of Fig1: G6PD should not be in italic. Same remark for Fig2.
Response 2:
Yes, we agree. The sentence was modified.
Comment #3: - p.4, line 11: This sentence is still unclear. “Phylogenetic analysis revealed the presence of three clades were G6PD sequence from GDI was clustered with the most closely related organisms such as Gluconacetobacter liquefaciens and Gluconacetobacter etanii (Figure 2)”.
Response 3:
Yes, we agree. The sentence was modified.
Comment #4: - Do the authors mean “where” instead of “were”? Could the sentence be simplified? “Phylogenetic analysis revealed that the G6PD sequence from GDI is most closely related to the G6PD sequences found in organisms such as Gluconacetobacter liquefaciens and Gluconacetobacter etanii (Figure 2).”
Response 4:
Yes, we agree. The sentence was modified.
Comment #5: - p.6, line 8. I would simplify as followed: “The G6PD protein, which is fused to an N-terminal His tag, was purified by Ni Sepharose High Performance affinity column (GE Healthcare). Then, the tag was removed using the site-specific protease HisTEVP”.
Response 5:
Yes, we agree. The sentence was modified.
Comment #6: - p.6, line 11: very long sentence. “The inset in figure 3 shows a single band (monomer) of the purified recombinant G6PD protein with an apparent MW of 56.4 kDa that was estimated by SDS-PAGE, which corresponds to the theoretical values previously reported and that is similar with those reported for other prokaryotes such as Thermotoga maritima and Pseudomonas aeruginosa [186,197], whereas it is different for the organs of mammals, such as the camel liver, rat kidney, and buffalo since they have MWs of around 60 to 75 kDa [2018-220].”
My sugegstion: “The inset in figure 3 shows that, after SDS-PAGE, the protein is detected as a single band with an apparent MW of 56.4 kDa. This is consistent with the theoretical MW of the monomeric form of G6PD. Of note, this size is similar to those reported for G6PD proteins in other prokaryotes such as Thermotoga maritima and Pseudomonas aeruginosa [186,197]. In mammals, such as the camel liver, rat kidney, and buffalo, MWs around 60 to 75 kDa have been documented [2018-220].”
Response 6:
Yes, we agree. The sentence was modified.
Comment #7: - p.6, line 22: “In addition, we found a second small peak with a relative MW of around 120 kDa, which could be a dimeric form, but no G6PD activity was observed for this form (Figure 3B). Furthermore, no monomer MW or larger aggregates species were observed in the Fast Protein Liquid Chromatography (FPLC) chromatogram.”
Response 7:
Yes, we agree. The sentence was modified.
Comment #8: - p.6, line 28: reference 24 refers to the rat protein, which has been removed from the sentence.
Response 8:
The reference was removed from Reference Section.
Comment #9: - p.7. line 6: “As can be observed at pH values ranging from 2.6 to 4.0, no G6PD activity was detected, while that at a pH value of 5.0, G6PD activity was found around 10%.”
Response 9:
Yes, we agree. The sentence was modified.
Comment #10: - p.7, line 10 “such as NADPH (notably required for lipid biosynthesis and for conferring protection against…)”
Response 10:
Yes, we agree. The sentence was modified.
Comment #11: - p.7, line 17: “ To investigate the effect of temperature on the activity of G6PD protein, aliquots of the enzyme were incubated at different temperatures (ranging from 15 to 60 °C),…”
Response 11:
Yes, we agree. The sentence was modified.
Comment #12: - p.7, line 19: please check the temperature values and sentence structure. Based on the graph, I would suggest writing: “the activity remained intact from 37 to 50°C, slowly decreased between 47 and 52°C, then sharply dropped. Furthermore, ……………….when incubated at temperatures below 30°C”
Response 12:
Yes, we agree. The sentence was modified.
Comment #13: - p.8, line 13: “As shown in Figure 6A, when the enzyme was incubated with 1 Km (7.2 μM) of NADP+, no protective effect on thermal stability was observed, since its temperature T50 (temperature at which the enzyme loses 50%) remained similar with or without NADP+. This result is in disagreement with that previously reported for recombinant human G6PD, whose T50 increases from 47°C to 59°C when in the presence of NADP+ (46).”
Response 13:
Yes, we agree. The Sentence was modified.
Comment #14: - p.9, line 7: “that has been reported to participate…”
Response 14:
Yes, we agree. The Sentence was modified.
Comment #15: - p.9, line 18. I would remove the first sentence and start directly by “We incubated G6PD with increasing concentrations of Gdn-HCl to induce protein unfolding. Then, we estimated the residual activity level of the enzyme, which is indicative of its folding state. Figure 6B shows a progressive loss of residual activity that is Gdn-HCl concentration-dependent, since…”.
Response 15:
Yes, we agree. The Sentence was modified.
Comment #16: - p.9, line 28 : “In addition, we also evaluated…”
Response 16:
Yes, we agree. The Sentence was modified.
Comment #17: - p9, line 30 “no changes were observed …”
Response 17:
The Sentence was modified.
Comment #18: - p.11, lines 17-22; The part starting by “As previously noticed…” is very difficult to follow. Please revise the structure of this paragraph. I suppose that the authors mean that tryptophan residues could be exposed to the environment after Gdn-HCl-induced unfolding, but the structure of the sentence is confusing. The point regarding exposure to high Gdn concentrations also needs to be clarified (concentrations >0.6 M) did have a high effect on the protein…
Response 18:
Yes, I agree with you. The sentence was removed because at high Gdn-HCl concentrations we observed a have a high effect on the protein .
Thank you very much for your observations.
Sincerely,
Dr. Saúl Gómez-Manzo
Laboratorio de Bioquímica Genética.
Torre de Investigación, 6º piso.
Av. Insurgentes sur 3700-C, México DF 04530.
Phone + (52 55) 10840900 ext. 1442
E-mail: saulmanzo@ciencias.unam.mx
Round 3
Reviewer 1 Report
I asked the Authors to provide the alignments, trees and model files. Although, the Authors claimed that they included them to Supplementary Data Files, currently I can download only one supplementary file with word document which is not what I wanted (maybe this is some bug of the manuscript submission platform, but I cannot get pdb file you mentioned in the response). This also means that I should apologize for not being clear enough.
a) When I requested the alignments, I asked for separate files (e.g., in Clustal, Nexus, Philip formats). Thus, I asked for the data underling Figure S1. Additionally, Table S2 is mislabeled as it does not have multiple sequence alignment, rather some (specific) data about protein sequences used in MSA.
b) When I requested the trees, I asked for separate files (e.g. in Newick or Nexus formats). Thus, I asked for the data underling Figure S2.
c) When I requested the bootstrap values, I asked for adding the bootstrap values for each branching point in the tree (e.g., let us consider the branching point of Homo sapiens vs. rodents (R. norveticus, M. musculus and C. griserous) - adding here the number, for instance, 98, coming from bootstrapping inform you that in 98% cases this branching occur after re-shuffling the data - thus it is well supported). This is the only reason you do bootstrapping. It is not enough to report only the bootstrapping total number as you did.
d) When I requested adding models, I asked for pdb file(s) that I can open in PyMOL, Jmol, VMD or USCF Chimera viewers. Thus, I asked for the data underling Figure S3.
In all cases (a-d), you can add the files as Supplementary Data files directly to the manuscript or (if not possible) store such files in the dedicated platforms (e.g., FigShare). When you do that, I (and the future readers) will be able to check the correctness of your analysis and conclusions (or use for future studies in case of the readers).
Author Response
October 17th, 2019.
Dear reviewer #1:
We are very grateful for your valuable observations that allow us to improve this manuscript
Comments and Suggestions for Authors
Comment #1: - I asked the Authors to provide the alignments, trees and model files. Although, the Authors claimed that they included them to Supplementary Data Files, currently I can download only one supplementary file with word document which is not what I wanted (maybe this is some bug of the manuscript submission platform, but I cannot get pdb file you mentioned in the response). This also means that I should apologize for not being clear enough.
Response 1:
Thank you very much for your valuable comments.
In our opinion, we have carry out all the comments and suggestions made for the reviewer 1. However, we have not correctly understood their observations because it was not being clear enough.
We thank the reviewer for point this out. We also respectfully asked the reviewer to consider that in the previously revisions, the reviewer initially suggested “The alignment(s), the tree(s), and the proteins model(s) should be added as supplementary material or stored in the external databases (e.g., FigShare)”. Based on their observations we decided to add the alignment(s), the tree(s), and the proteins model(s) as supplementary material. Unfortunately, due to some difficulties with the manuscript submission platform, it was not possible load the PDB file. Whereby, we are sending the files that are being requested by the reviewer in ZIP format.
Comment #2: - a) When I requested the alignments, I asked for separate files (e.g., in Clustal, Nexus, Philip formats). Thus, I asked for the data underling Figure S1. Additionally, Table S2 is mislabeled as it does not have multiple sequence alignment, rather some (specific) data about protein sequences used in MSA.
Response 2:
We are sending the file (Alignment ZWF.phy) with the data related with the Figure S1. This file was obtained with BioEdit program. The alignment was performed with ClustalW and was visualized with the online program Jalview (Figure S1). We attach the file Alignment ZWF.phy
Furthermore, we agree with you. The Table S2 was modified.
Comment #3: - b) When I requested the trees, I asked for separate files (e.g. in Newick or Nexus formats). Thus, I asked for the data underling Figure S2.
Response 3:
To generate the trees, we used the MEGA program. First we performed an alignment with ClustalW and the result was saved as ZWF CLUSTALW.mas format. Subsequently, ZWF CLUSTALW.mas was used to generate a phylogenetic tree and was constructed in MEGA7 (v. 7.0.21) with the Neighbor Joining method; then the file was saved as TREE ZWF.mts. However, the generated figure of the phylogenetic tree can only be saved in extension (.mts) using MEGA7 (v. 7.0.21).
Therefore we attach the files with the data related with the Figure S2 (ZWF CLUSTALW.mas and TREE ZWF.mts)
Comment #4: - c) When I requested the bootstrap values, I asked for adding the bootstrap values for each branching point in the tree (e.g., let us consider the branching point of Homo sapiens vs. rodents (R. norveticus, M. musculus and C. griserous) - adding here the number, for instance, 98, coming from bootstrapping inform you that in 98% cases this branching occur after re-shuffling the data - thus it is well supported). This is the only reason you do bootstrapping. It is not enough to report only the bootstrapping total number as you did.
Response 4:
We added the bootstrap values for each branching point in the tree. The Figure S2 was replaced in supplementary material file.
Comment #5: - d) When I requested adding models, I asked for pdb file(s) that I can open in PyMOL, Jmol, VMD or USCF Chimera viewers. Thus, I asked for the data underling Figure S3.
Response 5:
We are sending the pdb file (ZWF model). The ZWF model obtained is shown in Figure S3, panel A. The Figure S3, panel B was obtained with a structural alignment of the human G6PD enzyme (PDB entry 2BH9; gold), G6PD from L. mesenteroides (PDB entry 1H9A; indigo) and the minimized model of the G6PD from GDI (medium see green). Finally the Figure S3 panel C was obtained using the ZWF model and shows the representative residues (T37, P171, and K204) of G6PD active site.
Comment #6: - In all cases (a-d), you can add the files as Supplementary Data files directly to the manuscript or (if not possible) store such files in the dedicated platforms (e.g., FigShare). When you do that, I (and the future readers) will be able to check the correctness of your analysis and conclusions (or use for future studies in case of the readers).
Response 6:
We are added the files as Supplementary Data files directly to the manuscript. All the files were add as ZIP file.
Thank you very much for your observations.
Sincerely,
Dr. Saúl Gómez-Manzo
Laboratorio de Bioquímica Genética.
Torre de Investigación, 6º piso.
Av. Insurgentes sur 3700-C, México DF 04530.
Phone + (52 55) 10840900 ext. 1442
E-mail: saulmanzo@ciencias.unam.mx
Round 4
Reviewer 1 Report
There are only a few technical issues left (nevertheless, they should be addressed).
1) Please, add references to Table S2 in the main text. In the supplement you can also add information about supplementary files containing trees, alignments, etc.).
2) Remove typos, wrong case letters, and double spaces. For instance:
a) "Table S2. G6PD protein sequences used in the Multiple alignment.", the word "Multiple",
b) "Lane 2. Purified" to "Lane2: purified" and "Blue spot" into "blue spot"
c) "As see in Figure 4A" should be "as seen" or "as you can see"
d) For consistency "Acetobacter" should be italic.
...
and many others
At this point, it is up to you to find the remaining simple errors and fix them. I am not your proofreader, but it hurts to read the manuscript.
3) Rewrite:
a) "Finally, from solved 3D structure, we developed a model for the G6PD protein, which suggests the structural and functional features of G6PD from several organisms, including humans."
This statement is not precise and it suggests that you solved the 3D structure using e.g., the crystallography which is not the case. Write something like "... using homologous 3D structures we modeled the G6DP protein".
b) "multiple alignment was executed with ClustalW" - executed is a wrong word, use better one (e.g., done, performed, prepared, etc.)
c) "ClustalW using the online program Jalview [15] in order to find conserved sequences" in your case it should be "to find G6PD orthologs in GDI"
d) "three consensus sequences were identified" either there is one consensus sequence (the consensus is a unique entity otherwise it is not consensus) or three conserved fragments (I think this is what you wanted to write). Use "fragment" also later in the text, as it is more clear (by default when you say sequence you should think about the whole sequence of the protein, not the fragments). Moreover, such short fragments are commonly called motifs.
4) Supporting files format
a) I ask for specific formats (which are not binary formats and which are tool independent). Please do not use MEGA specific, binary formats such as .mas and .mts (MEGA session file cannot be open in anything than MEGA).
b) The provided tree file does not contain branch data from bootstrapping (fix it).
c) the numeration presented in Figure S1 is wrong - the fragment YIF... does not start from 100, either it is 12 (w/o gaps) or 49 (with gaps)
When the authors correct the mentioned issues, the manuscript will be ready, I will have no objection to publishing it.
General comments (do not need to be addressed):
- find a (better) bioinformatician to the next project
- use full stops more often instead commas and the text will be more readable straight away.
Author Response
"Please see the attachment"
